# The Importance of the One Health Concept in Combating Zoonoses

**DOI:** 10.3390/pathogens12080977

**Published:** 2023-07-26

**Authors:** Elina Horefti

**Affiliations:** Public Health Laboratories and Diagnostic Department, Hellenic Pasteur Institute, 11521 Athens, Greece; horefti@pasteur.gr; Tel.: +30-2106478819

**Keywords:** one health, zoonosis, zoonotic infections, outbreaks, epidemics

## Abstract

One Health fundamentally acknowledges that human health is linked to animal health and the environment. One of the pillars One Health is built on is zoonoses. Through the years, zoonotic infections have caused numerous outbreaks and pandemics, as well as millions of fatalities, with the COVID-19 pandemic being the latest one. Apart from the consequences to public health, zoonoses also affect society and the economy. Since its establishment, One Health has contributed significantly to the protection of humans, animals, and the environment, through preparedness, surveillance, and mitigation of such public dangers.

## 1. Introduction

According to the World Health Organization (W.H.O.), «One Health» is an interdisciplinary approach to studying public health issues in humans, animals, and their surrounding environments. This is accomplished through the implementation of specifically designed programs and research activities that involve multiple sectors [1].

One Health basically recognizes the fact that the well-being of humans is interchangeably connected with the well-being of animals and the environment, and vice versa. The action framework for «One Health» comprises of five distinct pillars: food safety, the bond formed between humans and animals, antimicrobial resistance, water contamination, and zoonoses. A basic prerequisite for the establishment and organization of a «One Health Platform» is the development of an interdisciplinary team that consists of public health experts, like medical doctors, veterinarians, biologists, and ecologists, as well as public health administrative officers. As far as zoonoses are concerned, the role of the aforementioned experts is the immediate response and intervention in the case of a zoonosis epidemic, as well as the surveillance and mitigation of such public threats.

A zoonosis can be accurately defined as any disease or infection that is naturally transmissible from vertebrate animals to humans and vice versa; their causative agents are fungi like *Microsporum* spp. [2], bacteria such as *Bartonella henselae* [3], viruses like West Nile and dengue viruses [4], parasites such as *Leishmania* [5] and *Toxoplasma gondii* [6], and prions such as the one responsible for transmissible spongiform encephalopathies [7]. 

Zoonoses can be classified into three distinct categories: (a) endemic zoonoses, which are widespread and affect both humans and animals, like *Brucella* and rabies virus [8], (b) epidemic zoonoses, which have sporadic temporal and spatial distributions, like the 2009 H1N1 influenza pandemic [9], and (c) emerging and re-emerging zoonoses, which have already existed but are now rapidly expanding in prevalence or range, like MERS [10]. With respect to their maintenance cycle, and according to Chomel, zoonoses can also be divided in four distinct groups: orthozoonoses or direct zoonoses, which are transmitted from an infected vertebrate host to another vertebrate through direct touch, fomites or a mechanical vector; cyclozoonoses, where more than one vertebrate species but no invertebrate host is required in order to complete the agent’s developmental cycle; pherozoonoses or metazoonoses, which need both vertebrates and invertebrates in order to complete their infectious cycle; and saprozoonoses that have both a vertebrate host and an inanimate developmental site or reservoir [11].

## 2. Emergence of Zoonotic Infections

As aforementioned, the causative agents of zoonoses are viruses, bacteria, fungi, parasites and prions. But, under which circumstances do these microorganisms emerge?

Firstly, the consumption of food and water contaminated with pathogenic bacteria and viruses can be a means of developing foodborne and water-borne zoonoses, respectively. According to the ECDC, and regarding foodborne outbreaks (FBOs), *Salmonella*, *Norovirus,* and *Campylobacter* were the zoonotic agents responsible for the highest number of outbreaks and cases in European countries [12]. In 2021, 113 confirmed cases of non-typhoid *Salmonella*, specifically *Salmonella* Braenderup, as well as 17 cases of Shiga toxin-producing *Escherichia coli* (STEC) were recorded in the UK; both infections were due to the consumption of Gaia melons imported from Latin America [13]. Another example of food-mediated zoonosis, this time on the Asian continent, was *Toxoplasma gondii* detected in the milk of camels, buffalos, and cows in the Iran province of East Azerbaijan [14]. This milk could have led to toxoplasmosis in humans if it had been consumed. 

Global warming and, consequently, climate change favor the emergence and re-emergence of zoonoses, because the changes in the dynamics of the hosts, the vectors and the pathogens, as well as their interactions, lead to significant changes in their epidemiology [15]. Climate change and rising temperatures encourage the development of zoonotic hosts, and more people are exposed to vector-borne diseases, because when temperatures rise, both pathogens and vectors reproduce more quickly [16]. This is the case with arboviruses, with a wide range of mosquito species being the primary vector for arboviral infections. For example, in coastal Kenya in 2004, the higher than usual temperatures, as well as the infrequent replenishment of home water supplies, may have aided the rise in Chikungunya virus infection [17]. Furthermore, it was recently shown, using the algorithm XGBoost, that there is a serious risk of an up to five-fold increase in West Nile virus outbreaks in the European region for the period 2040–2060 [18]. Finally, climate change was shown to increase the populations of bats in the areas of Yunnan province in China and neighboring Laos and Myanmar, which resulted in approximately 100 bat-borne coronaviruses in these regions, thus enabling the emergence of SARS-CoV-1 and SARS-CoV-2 [19]. 

The human factor also plays a significant part in the emergence and re-emergence of zoonoses. Alterations in the landscape for economic and social reasons change the ecosystem and cause stress to wildlife species which move to urban areas; if these species carry a zoonosis, as a vector or an intermediary host, it may be just a matter of the zoonosis being spilled over to humans [20]. Deforestation, accurately defined as the « destruction or removal of forests and their undergrowth», can result in crucial changes to biodiversity [21,22]. Finally, pollution, water management, putrescible waste management, vector ecology, urban microclimates, and human encroachment on wildlife habitats, all have an impact on infectious diseases at the human–wildlife–domestic animal interface [23].

Globalization is, according to the W.H.O., the “increased interconnectedness and interdependence of peoples and countries. It is generally understood to include two inter-related elements: the opening of international borders to increasingly fast flows of goods, services, finance, people and ideas; and the changes in institutions and policies at national and international levels that facilitate or promote such flows” [24]. Apart from all the advantages that can benefit humanity from globalization, there are downsides as well. An example is the increase in *Trichinella* spp., which affect swine, pork meat, and, subsequently, humans, which is a food-borne zoonosis shown to rise due to illegal meat transportation and the introduction of new eating habits [25]. 

During the last years, due to political instability worldwide, there has been a significant rise in immigrants and refugees. According to the United Nations, “89.3 million people were forcibly displaced world-wide at the end of 2021. Among those, 27.1 million were refugees. There were also 53.2 million internally displaced people, 4.6 million asylum seekers, and 4.4 million Venezuelans displaced abroad” [26]. The population movement, paired with various social factors, such as the length of stay in host areas, disease exposure, lifestyle changes, lack of healthy practices, and limited access to healthcare can lead to zoonotic epidemics [27]. 

Natural phenomena and events, such as heavy rainfall and floods, earthquakes or tsunamis, can also be the cause of the emergence or re-emergence of zoonoses [28]. For example, hantavirus and leptospirosis usually emerge after heavy rain and floods [29]. Vector-borne diseases are very likely to emerge after heavy rain, as in the case of Hurricane Michelle in 2001 in Cuba, which probably resulted in the initiation of a dengue virus epidemic [30]. 

Earthquakes may result in landslides or other environmental modifications, disturb animal habitats and change the distribution of species, thus leading to a more extensive spread of zoonotic infections [31]. Haiti, in 2010, had an ongoing cholera epidemic; after the earthquake that took place in the same year, the serious inadequacies in the infrastructure for animal health and the water supply that already existed, not only worsened the cholera epidemic, but also led to the spread of other zoonotic infections, like malaria and dengue fever, due to a change in the mosquitoes’ lifecycles [32]. 

A lack of communication between veterinarians, physicians, and wildlife experts also plays a major role in extensive zoonotic infections. A characteristic example of such a lack of communication is the case of the Nipah virus in Malaysia, in 1998. The virus was initially transmitted from bats to pigs through the consumption of fruits and, finally, from pigs to humans. The result was 265 cases of acute encephalitis and 105 deaths [33]. 

## 3. The Consequences

The most important impact of zoonoses is on human health and, almost a decade ago, it was shown that they are responsible for almost 2.5 billion cases of human infection and approximately 2.7 million human deaths globally, every year [34]. 

Apart from the cost of human lives and the problems caused to public health, zoonoses have significant socioeconomic consequences on everyday life, both in terms of direct costs associated with treating infected individuals and indirect costs associated with lost productivity, reduced trade, and increased public health measures. 

This is the case with highly pathogenic avian influenza strains, which can kill up to 90–100% of the flock, immediately causing economic losses. It can also lead to trade restrictions on animals and animal products, which can reduce the profitability of the agricultural sector. Also, in livestock, illness may cause a reduction in productivity, as well as in meat and milk production [35]. A loss of productivity also occurs when people who are infected by zoonotic diseases are not able to work or perform their usual activities. This, in turn, results in lost income and reduced economic output.

Zoonotic diseases often require expensive medical treatments, which can be a significant burden on individuals, families, and healthcare systems. In a piece of research conducted in six African countries in 2016, the economic burden on human health due to cysticercosis, a tissue infection caused by a young form of the pork tapeworm [36], was estimated to be approximately USD 241 million purchasing power parities [37]. 

Public health measures need to be implemented by governments, in order to control the spread of zoonotic diseases, such as quarantines, travel restrictions, and vaccination campaigns. New York City, for example, invested nearly USD 2.44 billion in direct costs into the vaccine campaign against COVID-19 between 14 December 2020 and 31 January 2022 [38]. 

Zoonotic diseases can also affect the tourism industry by reducing the number of visitors to affected areas and causing financial damage to businesses that rely on tourism. It was shown in the past how tourism was reduced in countries with zoonotic infections, as in the case of Singapore in 2004 due to SARS [39] and Mexico at the beginning of the H1N1 influenza pandemic [40].

Overall, zoonotic diseases can have significant economic consequences, which can be reduced by investing in surveillance, early detection, and rapid response systems to prevent the spread of these diseases.

## 4. Zoonoses in Time—From Ancient Greece to Modern China and Europe

The history of humanity is filled with great explorations, discoveries, innovations, and life changing scientific and cultural breakthroughs; it is also full of untimely and unfair deaths caused not only by wars, but by deadly infections and diseases as well (Table 1). Unfortunately, SARS-CoV-2 is not the first zoonosis in the history of humanity which has caused a pandemic with more than 767 million confirmed cases and almost 7 million deaths (up to 7 June) [41].

The first reported zoonotic disease in history was the plague in Athens in 430 BC, which killed almost 100,000 people with statesman Pericles, the founder of democracy, one of its most “famous” victims [42]. Contemporary physicians and epidemiologists evaluated the symptoms reported by the historian from this period, Thucydides, and suggested that it was due to one of the following infectious agents: smallpox, typhus or measles [43]. It was also speculated that this infectious disease had actually spread from the African continent to Athens. The extensive research on the matter concluded that the zoonosis was most probably typhoid fever, but this was later contradicted [44,45]. 

The Justinian plague in 541 AD, as well as the “Black Death” in 1345, were two pandemics caused by *Yersinia pestis* [46]. In both cases, almost 50 million people died. Another zoonotic epidemic was the “American Plague”, which lasted from 1793 to 1798, and killed almost 25,000 people; it was due to yellow fever virus transmitted through the mosquito *Aedes aegyptae* and it occurred through the arrival of almost 3000 refugees from Saint-Domingue (now Haiti) to the port of Philadelphia [47]. 

Since the beginning of the 20th century, many zoonotic diseases have emerged. First was the “Spanish Flu”, which lasted from 1918 to 1922, and claimed the lives of more than 20 million people. It was Influenza Type A and, more specifically, H1N1, which was later shown to have originated in birds [48]. In 1957, what is now known as “Asian Flu” occurred, which originated in Guizhou in southern China and later spread in Hong Kong, Singapore, Japan, the United States of America, the United Kingdom, and Germany, causing 1.1 million deaths [49]. The strain that caused the epidemic was H2N2 and although it was expected to last for a short period of time, it finally disappeared after 11 years [50]. In 1968, the H2N2 strain was replaced by the H3N2 Influenza strain, originating from Hong Kong, and later spreading to the rest of the world, with a global estimate of 1 million deaths [49]. 

In 1976, a new virus emerged in Africa, which caused severe hemorrhagic fever: Ebola. Two epidemics were reported at the same time, one in the Democratic Republic of Congo and one in Sudan [51]. Since then, Congo has had more than ten Ebola outbreaks, especially the one in 2018, being the harshest, with more than 3470 confirmed cases and 2287 fatalities. The virus re-emerged in 2020 and continued, with the last confirmed cases reported in 2022 [52]. 

In 1980, the human immunodeficiency virus emerged in Central Africa, when humans came in contact with the blood of chimpanzees. A new pandemic occurred, and unfortunately it is still going on. The first official reporting of what would be later known as AIDS (acquired immune deficiency syndrome) was in 1981, described as an infection with *Pneumocystis carinii* in five young, otherwise healthy, homosexual men [53]. It was years later when the infection was scientifically linked to animals, and it was fully explained how it occurred [54]. 

Unfortunately, the historical retrospective on zoonoses does not stop with HIV. In 1996, avian influenza H5N1 made its first appearance in domestic waterfowl in southern China and continued its spread in 1997, with poultry outbreaks both in China and Hong Kong. As far as humans are concerned, this year there were 18 confirmed cases and 6 deaths in Hong Kong. H5N1 re-emerged in 2003 in China and spread to other Asian countries as well [55]. Since 1997 and up to 2015, 907 human confirmed cases of avian influenza H5N1 have been reported globally, as the virus spread from eastern Asia to Africa, with case fatalities equal to almost 50% [56]. A H7N9 avian influenza was first reported in China in 2013 [57] and an epidemic occurred in 2016; 1220 laboratory confirmed cases and 494 deaths were reported [58]. Staying on the influenza topic, in 2009 a novel influenza strain, H1N1, was the cause of the first pandemic in 100 years, resulting in 151,700 to 575,400 deaths worldwide [59]. 

Zika virus first emerged in 1947 in Uganda, and since the 1950s sporadic infections by the virus have been identified in many African countries. In 2015, Zika virus caused a large epidemic in Brazil and an association between confirmed Zika virus cases in pregnant women and the outbreak caused a «Public Health Emergency of International Concern» [60,61]. Since 2017, though, the cases have declined significantly. 

Influenza viruses and arboviruses are not the only types of zoonoses that cause epidemics and pandemics, as coronaviruses have played a major role in the emergence of zoonotic diseases. Firstly, there was the SARS outbreak, which lasted for two years, specifically from 2002 to 2004. The viral strain was SARS-CoV-1, which belonged to the *Betacoronavirus* genus. The outbreak was initially identified in Guangdong, a province in south China close to Hong Kong, and resulted in 8,110 confirmed cases and 811 deaths in 30 territories in China, Europe, and North America [62]. The virus spread from bats, the original host, to humans through an intermediate host, which in this case was civets [63]. 

MERS followed, as a zoonosis of coronaviral origin, in 2012 in Saudi Arabia. Once again, bats were the original host of the virus, which spilled over into human populations through livestock, specifically camels [64]. As of January 2020, the confirmed cases of MERS were 2519 with 866 deaths and a 34.3% fatality rate [65]. 

Last, but certainly not least, in the list of the zoonoses caused by coronaviruses, is what was originally known as the Wuhan virus or 2019-n-CoV, finally named SARS-CoV-2, which caused one of the largest pandemics in history. The emergence of SARS-CoV-2 is linked to a complex interplay of factors, including the close interaction of humans and animals, international travel, and global trade [66]. The virus is thought to have originated in bats and may have been transmitted to humans through an intermediate animal host, possibly a pangolin. It then spread rapidly among humans through person-to-person transmission. 

In the shadow of COVID-19, the monkeypox virus, an infection common in Central and West Africa, re-emerged. The first outbreak was recorded in 1958 [67] and many cases have been recorded since, in many African countries like Nigeria in 2017 [68]. The virus re-emerged in 2022 in London, from an imported case from Nigeria, where the disease is still endemic [69]. Overall, 87,942 cases have been recorded worldwide with 146 deaths so far [70]. 

Currently, according to the ECDC, in 24 countries around Europe, between 3 December 2022 and 1 March 2023, 522 domestic and 1138 wild birds were reported to have contracted the highly pathogenic avian influenza A(H5N1) virus. Surprisingly enough, the HPAI virus was discovered to have infected a large number of marine birds as well, specifically black-headed gulls. So far, though, it has been assessed by experts that the risk of infection by avian H5 influenza viruses currently circulating in Europe is considered low for the general population in the EU/EEA and low-to-moderate for those exposed through occupational or other exposures [71]. 

## 5. The One Health Concept

The One Health concept was initially introduced in 1850, when Rudolph Virchow first used the term “One Medicine” in order to define the lifecycle of *Trichinella spiralis*. Virchow was actually the first scientist who described how *Trichinella spiralis* affects both humans and animals, particularly swine, leading to the “discovery” of zoonoses and how all living organisms are linked [72]. 

In modern times, the first time that health experts realized that zoonoses pose a great danger for both humans and animals was in 1997, in Hong Kong, when an avian influenza H5N1 outbreak led to serious losses of poultry, which were actually killed in order to protect public health, resulting in massive economic losses. On top of that, it was the first time that an infection from poultry infected humans, with 18 confirmed cases and 6 fatalities [55].

In August 1999, an outbreak of West Nile virus occurred in New York; it was actually the first time this virus had been detected in the western Hemisphere [73]. Local health officials noticed an increase in bird fatalities in New York City, mainly crows, before and during this outbreak. One month later, it was found with real-time PCR and sequencing that the pathogenic agent responsible for both the fatalities amongst birds and human meningoencephalitis cases in humans was West Nile virus [74], solid proof that animal and human health are interconnected. 

From 1997 to 2003, H5N1 viruses were not widely detected. In 2003, though, they reemerged in China and other nations, causing extensive poultry outbreaks throughout Asia. Two years later, in 2005, H5N1-positive wild birds infected poultry across Europe, the Middle East, and Africa [75], resulting in a serious economic and ecological disaster. 

In parallel with the H5N1 cases, SARS virus emerged in China. Public health experts, microbiologists, and epidemiologists around the globe realized that a previously unknown pathogen could emerge from a wildlife source at any time and in any place and, without warning, threaten the health, welfare, and economies of entire societies. Also, there was a clear need for all countries to be able to maintain an effective “alert system”, in order to quickly detect potential outbreaks, as well as to share information about such outbreaks transparently. Finally, they all came to the conclusion that a quick and efficient response to pandemics requires global collaboration, providing the framework for the establishment of the concept of One Health.

In 2007, a conference, «The International Ministerial Conference on Avian and Pandemic Influenza» took place in New Delhi, with delegates from 111 nations and 29 international organizations. As recommended by this conference, the FAO, WOAH, WHO, UNICEF, the World Bank, and the United Nations System Influenza Coordination (UNSIC) collaborated in order to create a document titled “Contributing to One World, One Health^TM^-A Strategic Framework for Reducing Risks of Infectious Diseases at the Animal-Human-Ecosystems Interface” [76]. One Health was officially established. 

## 6. The Success Stories

There are numerous cases where One Health has succeeded in mitigating zoonotic infections. An example is the situation of rabies in Sri Lanka, Bhutan, and Bangladesh, where the infections were uncontrollable. Health experts and officials followed the proposed One Health methods in order to mitigate the infections and deaths, such as surveillance, outbreak investigation, laboratory testing, the strategic mass vaccination of dogs, and human vaccinations and dog population management. At the same time, professional training and cross-sectoral collaboration were reinforced. In almost ten years, the confirmed cases of rabies in Bhutan were only 17 from 2006 to 2016, in Bangladesh human fatalities decreased from 1500 to 200 from 2012 to 2015, and in Sri Lanka there were less than 50 deaths in 2012 [77]. 

Hendra virus is transmitted through bats to horses and then to humans, and has a fatality rate equal to 60% in humans and 75% in horses [78]. The virus was first detected in Brisbane, Australia, in 1994, when 21 horses were infected; two people were also infected with one fatality amongst them [79]. From 1995 to 2023, almost 100 cases of Hendra virus in horses were reported, which all died [80], whereas only 7 cases were recorded amongst humans [81]. The mitigation of Hendra was achieved through the development and evaluation of a vaccine for horses [82], as well as a Hendra surveillance system in horses, empowered by the Australian Government.

In the case of MERS-CoV, the One Health approach was a valuable asset for the preparedness against the virus and its mitigation in Arab countries. In 2015, a workgroup took place in Qatar in collaboration with the FAO, WHO and OIE; representatives from Middle Eastern and Arab countries, as well as health experts from around the world, attended the workshop, in which policies and strategic plans were designed and implemented by veterinarians and public health experts [83]. According to the ECDC, «as of 7 June 2023, no MERS-CoV cases have been reported with date of onset in 2023 by health authorities worldwide or by WHO» [84]. 

Zika virus was thought to be a mild disease up until October 2015, when maternity facilities in northeast Brazil noticed a significant rise in the number of newborns born with microcephaly, an uncommon disorder linked to inadequate brain development; microcephaly was linked by health experts to Zika virus infection during pregnancy [61]. Up to December 2016, there were 174,667 confirmed cases, 528,157 suspected cases, and 18 deaths [60]. From February through to November 2016, the W.H.O. declared Zika virus infection «a public health emergency of global concern». Zika virus cases decreased globally starting in 2017 [85]. The outbreak was successfully controlled by basically implementing the One Health concept: mosquito control, vaccine development, diagnostic tests, and public campaigns [86,87].

Finally, DPT, namely diphtheria/tetanus/pertussis, and polio vaccination efforts for children in Chad and contagious bovine pleuropneumonia control for animals led to better coverage in both humans and livestock, and pastoralists were more conscious of public health issues. Also, vaccination and sterilization policies for dogs in Jaipur, India, caused a significant reduction in human rabies cases to zero (as opposed to instances rising in other states without similar efforts). The number of stray dogs has decreased by 28%. In Kyrgyzstan, veterinary and public health professionals started visiting farms together, thus succeeding in lowering the cost of brucellosis, echinococcosis, and other zoonotic disease surveillance. Finally, the integration of facilities for both animal and human health in Canada resulted in a 26% decrease in operating expenses [88]. 

## 7. The COVID-19 Case

The emergence of SARS-CoV-2 is the perfect example of a zoonosis-based pandemic. The virus is a zoonosis, a modified coronavirus of bat origin, which probably used pangolins as an intermediary host [89], originated in the Huanan Seafood Wholesale Market in Wuhan and started spreading amongst humans [90]. The W.H.O. declared the COVID-19 outbreak «a Public Health Emergency of International Concern» in January 2020, in the International Health Regulations Emergency Committee in Geneva, where 15 public health experts from the six regions were in attendance. In this meeting, the importance of the investigation on the source of the animals from which the outbreak began, as well as the extent of the human-to-human transmission, were underlined. Also, the control of the infection in other provinces of China, the strengthening of the surveillance for severe acute respiratory infections in these regions, and the strengthening of the containment measures were discussed. In true One Health style, the Committee made perfectly clear that providing information to the international community is essential to understanding the situation and its potential impact on public health [91]. 

After all these years of dealing with zoonoses, epidemics, and pandemics, and implementing the One Health approach successfully, one could actually have expected SARS-CoV-2 to be mitigated from the beginning. Clearly, this did not happen. If we find out why, then we might be more prepared in the future. 

One of the reasons is the origin of the zoonosis itself: wet markets are quite a common place of trade in Asian countries. This is where domestic, wild, and exotic animals coexist and are traded, with fresh meat is sold next to them, and thousands of people visit them every day. The interspecies transmission of viruses is something that has happened in the past with avian influenza [92] and possibly with SARS-CoV-2 [93]. 

The timing of the SARS-CoV-2 emergence was rather unfortunate, as it happened at the beginning of December 2019. At this time, there were many influenza cases in the region of Wuhan, as in many countries in the northern Hemisphere. The clinical symptoms of SARS-CoV-2 were easily classified as influenza without further laboratory confirmation. Also, many cases were asymptomatic and people who appeared to be healthy were actually spreading the virus. 

Globalization and international travel were also the reasons for the spread of SARS-CoV-2. Many visitors from Asian countries were in Milan in February 2020, or even in January 2020, and the spread of the virus was inevitable [94]. 

Investigations into SARS-CoV-2 were, and still are, heavily weighted toward tackling zoonotic diseases at the animal–human interface. However, very few researchers have mentioned interactions at the human–animal–environment interface [95], an oversight of great importance as the health of animals and humans is interconnected with the environment. The scientific community realized the hard way, not only that human and animal health are interconnected, but that the monitoring of every environmental change is crucial [96]. 

Finally, as it was stated by doctors working in the Papa Giovani XXIII Hospital in Bergamo «Western health care systems are patient-centered, but an epidemic requires a community-centered shift in perspective. Pandemic solutions are needed for the whole population, not just hospitals» [97]. 

## 8. Future Pathogens and Outbreaks

Recently, there has been much discussion about «Pathogen X», the unknown infectious disease that will cause a new pandemic. The W.H.O., though, has been preparing for such an incident since 2015, right after the Zika virus outbreak, when during a meeting in December of the same year, health experts concluded that there were seven diseases that required immediate R&D: Crimean-Congo hemorrhagic fever, filovirus diseases, highly pathogenic emerging coronaviruses relevant to humans, Lassa fever, Nipah virus, Rift Valley fever, and «preparedness for a new disease» [98]. The concept was to raise awareness, develop novel diagnostics, vaccines, and therapeutics, implement behavioral interventions, and fill critical gaps in scientific knowledge. The conclusions from this meeting were actually used a few years later, in order to help nations control COVID-19 with quick vaccine development and the application of restriction measures. 

A W.H.O. meeting about the unknown «Pathogen X» was held in August 2022 in Geneva, where the strategies that helped mitigate COVID-19, as well as the preparedness for the next pandemic, were discussed [99]. It was highlighted that the One Health concept must be followed in order to be prepared for pathogen emergence and address any potential health threat globally. 

## 9. Conclusions

The One Health concept recognizes the interconnectedness of human, animal, and environmental health. It aims to promote collaboration and communication across different sectors to address health risks that arise at the intersection of these domains. While One Health is an important approach for addressing emerging infectious diseases, it alone cannot prevent the emergence of new diseases. To prevent future pandemics, a multifaceted approach is needed that includes One Health, as well as improved surveillance, rapid response capabilities, and better communication and coordination among public health authorities at a global level. In simple words, what is needed is unity, teamwork, and transparency. 

## Figures and Tables

**Table 1 pathogens-12-00977-t001:** Timetable of major epidemics and pandemics caused by zoonoses.

Year of Emergence	Location	Zoonosis	Fatalities
430 BC	Ancient Greece	*Rickettsia prowazekii*?—“Plague”	100,000
541 AD	Byzantine Empire	*Yersinia pestis* —“Justinian Plague”	15–100 million ^(a)^
1345	Western Eurasia—North Africa	*Yersinia pestis*—“Black Death”	75–200 million
1793	Philadelphia, USA	Yellow Fever Virus	25,000
1918	Europe	Influenza Type A (H1N1 strain) “Spanish Flu”	20 million
1957	Asia, USA, Europe	Influenza Type A (H2N2 strain) “Asian Flu”	1.1 million
1968	Hong-Kong—Worldwide	Influenza Type A—H3N2 strain	1 million
1976	Democratic Republic of Congo and Sudan	Ebola Virus	280
1980	Central Africa—Worldwide	HIV—AIDS	40.1 million
1996	China—Hong-Kong	Influenza Type A (H5N1 strain) “Avian Influenza”	6
2002	South China, Europe, North America	SARS	811
2009	Worldwide	Influenza Type A (H1N1 strain) “Swine Influenza”	151,700–575,400
2012	Saudi Arabia	MERS	866 ^(b)^
2015	Brazil	Zika Virus	18
2016	China	H7N9 Influenza Type A (H7N9 strain)	494
2018	Democratic Republic of Congo	Ebola	2287 ^(c)^
2019	China	SARS-CoV-2	7 million
2022	London ^(d)^	Monkeypox	146

^(a)^ In a two century span, ^(b)^ as of January 2020, ^(c)^ re-emergence in 2020 and 2022, ^(d)^ imported case from Nigeria.

## Data Availability

All data supporting this review are available.

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
