# Peer review of "The Importance of the One Health Concept in Combating Zoonoses"

_pathogens, 2023, doi:10.3390/pathogens12080977_

Round 1

Reviewer 1 Report

The authors set to out to describe the importance of the one health concept in responding to zoonoses. The authors do an adequate job in this regard though they do not break any new ground. What I found particularly lacking in this review was figures and tables. The authors should summarize some of the data about zoonoses presented in the "zoonoses in time" section, for example, in a table. This would make the paper much more impactful and readable. Similarly, the "SUCCESS STORIES" section, and the "FUTURE PATHOGENS" section could be improved by the addition of tables and figures.

Line 116: not clear what "PPP" represents.

The manuscript will benefit from an English language editor. For example the word "zoonose" used extensively should be zoonosis. 

Author Response

Hello!

Firstly, I would like to thank you for your useful comments and suggestions.

I edited the manuscript and corrected the grammar and syntax errors. Also, I included a table for the timeline of zoonoses as suggested. All changes are marked in red color.

I hope that I addressed the issues you pointed out.

Kind regards,

Elina Horefti

Reviewer 2 Report

With this review, Dr. Elina Horefti gives an overview of the main topics concerning zoonoses, such as the historical and geographical origin, the different types of zoonoses and who or what they impact, the past outbreaks and also the potential new ones, focusing on the one-health approach to managing this public health issue, also indicating the ways in which pandemics can be prevented. This study may also be of interest to specialists worldwide . Please find more specific comments below.

Line 33 Cryptosporidium parvum is not classified as fungi but is included in the protozoa.

Please also describe the primary epidemiological classification based on the maintenance cycle of zoonoses. This classification is important when considering the various alternatives for control measures.

This classification is well described by:Chomel BB. Zoonoses. Encyclopedia of Microbiology. 2009:820–9. doi: 10.1016/B978-012373944-5.00213-3. Epub 2009 Feb 17. PMCID: PMC7149995.

Line 45-46, I suggest using the terms foodborne zoonoses and water-borne zoonoses to make this topic more transparent.

Line 52 uses foodborne zoonoses instead of food-mediated zoonoses

Line 72

In my opinion the authors should better explore zoonotic parasites and the  anthropogenic factors  (changing migration flows, globalization of food markets, pollution, etc.) involved in the spread of pathogens. These topics should be mentioned and described in this paragraph, therefore, I recommend taking into consideration the following papers:

Pozio, E. (2020). How globalization and climate change could affect foodborne parasites. Experimental Parasitology208, 107807.

Robertson, L.J., Sprong, H., Ortega, Y.R., Giessen, J.W.B. van der, and Fayer, R. (2014). Impacts of globalization on foodborne parasites. Trends in Parasitology, 30 (1), 37–52.

Alonso Aguirre, Changing Patterns of Emerging Zoonotic Diseases in Wildlife, Domestic Animals, and Humans Linked to Biodiversity Loss and Globalization, ILAR Journal, Volume 58, Issue 3, 2017, Pages 315–318, https://doi.org/10.1093/ilar/ilx035

 I also recommend to consult further papers on these topics

Line 51 the UK

Line 55 if it had been consumed

Line 66 of aN up to

Line 112 expensive medical treatmentS

Line 138 430 bc,

Line 143 HAD spread

Line 170 it’s still going on

Line 191 the cases have declined 

Author Response

Hello!

Firstly, I would like to thank you for your useful comments and suggestions.

I addressed the issues you pointed out and included new references. All changes are marked in red color.

Kind regards,

Elina Horefti